# E-CORE: Emotion Correlation Enhanced Empathetic Dialogue Generation

**Fengyi Fu[1], Lei Zhang[1]\*, Quan Wang[2], Zhendong Mao[1]**
[1]University of Science and Technology of China
[2]Beijing University of Posts and Telecommunications
`ff142536f@mail.ustc.edu.cn, leizh23@ustc.edu.cn,`
`wangquan@bupt.edu.cn, zdmao@ustc.edu.cn`

## Abstract

Achieving empathy is a crucial step toward humanized dialogue systems. Current approaches for empathetic dialogue generation mainly perceive an emotional label to generate an empathetic response conditioned on it, which simply treat emotions independently, but ignore the intrinsic emotion correlation in dialogues, resulting in inaccurate emotion perception and unsuitable response generation. In this paper, we propose a novel emotion correlation enhanced empathetic dialogue generation framework, which comprehensively realizes emotion correlation learning, utilization, and supervising. Specifically, a multi-resolution emotion graph is devised to capture context-based emotion interactions from different resolutions, further modeling emotion correlation. Then we propose an emotion correlation enhanced decoder, with a novel correlation-aware aggregation and soft/hard strategy, respectively improving the emotion perception and response generation. Experimental results on the benchmark dataset demonstrate the superiority of our model in both empathetic perception and expression.

## 1 Introduction

Empathy is a desirable human trait that improves the emotional perceptivity in emotion-bonding social activities, helping to achieve a humanized dialogue system (Smith, 2006; Singer and Lamm, 2009). Empathetic dialogue generation (EmpDG) which aims at perceiving the emotional expressions in dialogue to generate appropriate responses rich in empathy, is proposed and has attracted extensive attention with its ability to improve user experience and satisfaction in multiple domains (Fitzpatrick et al., 2017; Wang et al., 2021).

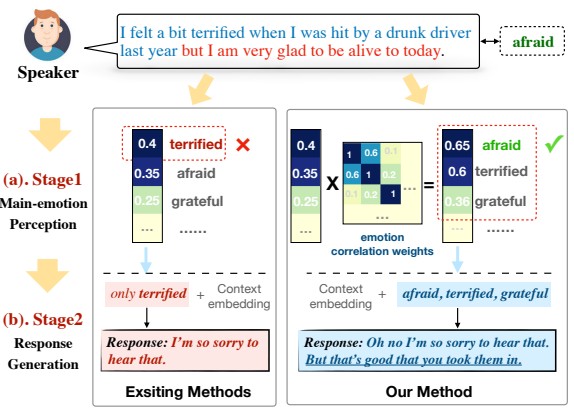

Figure 1: A real empathetic dialogue generation case based on our method (right) and existing methods (left), which is divided into two stages: (a) main-emotion perception, (b) response generation. For more detailed visualization of (a) refer to Fig.6.

Most existing methods follow a multi-task learning paradigm, jointly training an emotional classification task and dialogue generation task to achieve response generation with empathetic constraints. Recent works take their effort on two aspects. The first focuses on improving the emotion perception, for example, by introducing external knowledge (Li et al., 2020b, 2022b; Sabour et al., 2022), mining emotion causes (Kim et al., 2021; Gao et al., 2021), or more fine-grained emotion modeling (Li et al., 2020a; Kim et al., 2022). The other focuses on promoting the generation strategy, based on mixture of experts (Lin et al., 2019), different emotion look-ahead reward functions (Shin et al., 2020), emotional mimicry (Majumder et al., 2020) and so on. In general, these methods first perform main-emotion prediction with a single-label emotional classifier, then inject the predicted emotion into generation to achieve empathetic expression.

The above paradigm implicitly introduces an *independent assumption* on different emotions, both in modeling and utilization, which respectively from the learning for maximizing separation be-

---

\*Corresponding author. This work is supported by the National Key Research and Development Program of China under Grant (No.2021YFF0901600), the National Science Fund for Excellent Young Scholars under Grant (No.62222212), and the National Natural Science Foundation of China under Grant (No.61876223).

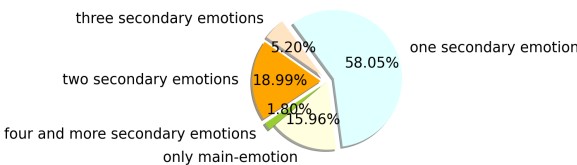

Figure 2: Statistics of secondary emotions proportion in the EMPATHETICDIALOGUES dataset samples.

tween different emotions in classification and the abandonment of secondary emotions in generation. However, studies on social psychology (Vansteelandt et al., 2005; Martinent et al., 2012) suggest that human emotions are not completely independent, but with an *intrinsic correlation*, manifested as dialogues and responses are typically accompanied with the *co-occurrence* of multiple emotions (Martinent et al., 2012). This independent assumption with ignoring the emotion correlation directly impairs main-emotion perception, as the main-emotion as a whole feature should be co-determined by the occurred emotions in context. (Fig.1-a, the common correlation weights of *grateful* and *terrified* helps distinguish the true emotions *afraid*). Moreover, this assumption is also harmful for response generation, as the model dominated by one emotion lacks the ability to recognize emotional transitions (Fig.1, a transition from *afraid* for accident to *grateful* for survival), resulting in unsuitable responses (Fig.1-b, only "sorry to hear" for survival). Therefore, considering the emotion correlation is necessary for precise emotion perception and better empathetic expression. Statistical result[1] on benchmark dataset in Fig.2, which is calculated based on the quantity of other-emotion-related words in samples, further suggests that the emotion correlation learning is significant for EmpDG task with a proportion of samples containing secondary emotions reaches **84.04%**.

As the annotation for all subtle emotions in dialogues is hard and inefficiency, we propose to mine and incorporate this *intrinsic emotion correlation* into single-labeled EmpDG. There are three challenges: 1) modeling and **learning** the multi-emotion correlation; 2) **utilizing** the correlated co-occurrence emotions without biasing toward to the labeled emotion; 3) providing **supervision** to avoid excessive or erroneous introduction of multi-emotion information.

To this end, we propose a novel **E**motion **COR**relation **E**nhanced empathetic dialogue gen-

---

[1]Specific details for statistics are supplied in appendix A.

eration framework, namly E-CORE, with three tailored modules to address above challenges. Specifically, we propose a novel directed weighted graph, which captures the subtle emotion interactions in context from different resolutions, further encoding the intrinsic emotion correlation. Then we design an emotion correlation enhanced decoder, which adopts a correlation-aware aggregation and a soft/hard strategy, incorporating the correlated co-occurrence emotions to improve emotion perception and response generation, respectively. Meanwhile, an emotion correlation loss is constructed to provide multi-emotion regular constraints.

Our contributions are summarized as follows: 1) We propose breaking the *emotion independence assumption* existing in current methods and modeling the *intrinsic emotion correlation*. To the best of our knowledge, this is one of the first frameworks in EmpDG that explicitly models and utilizes emotion correlation to enhance emotion perception and response generation. 2) We propose a distinctive method with three tailored modules respectively addressing the emotion correlation learning, utilizing, and supervising, which effectively and accurately capture the correlated co-occurrence emotions in dialogues even under single-label, enhancing empathy perception and expression. 3) Extensive experiments verify the superiority of our method on both emotion prediction (**8.34%** in accuracy) and response generation (**8.53%** in perplexity). Ablation studies and specialized experiments on constructed multi-emotion annotated sub-dataset also validates the fidelity of our emotion correlation learning.

## 2 Related Work

### 2.1 Emotional Dialogue Generation

In recent years, open-domain dialogue systems have achieved great progress (Li et al., 2016; Liu et al., 2016; Zhong et al., 2019; Zhang et al., 2020a; Shen et al., 2021; Zhu et al., 2022). As the combination of emotion and personality leads to a more human-like system, the emotional dialogue generation task which aims to generate emotional responses according to specified emotion label, was proposed and developed (Song et al., 2019; Dong et al., 2021; Ide and Kawahara, 2021; Liang et al., 2021; Li et al., 2021; Tu et al., 2022). Some works (Firdaus et al., 2021) also make efforts in multi-emotion guided generation, however, these works based on manually annotated emotions, focus on the encoding for provided multi-emotion.

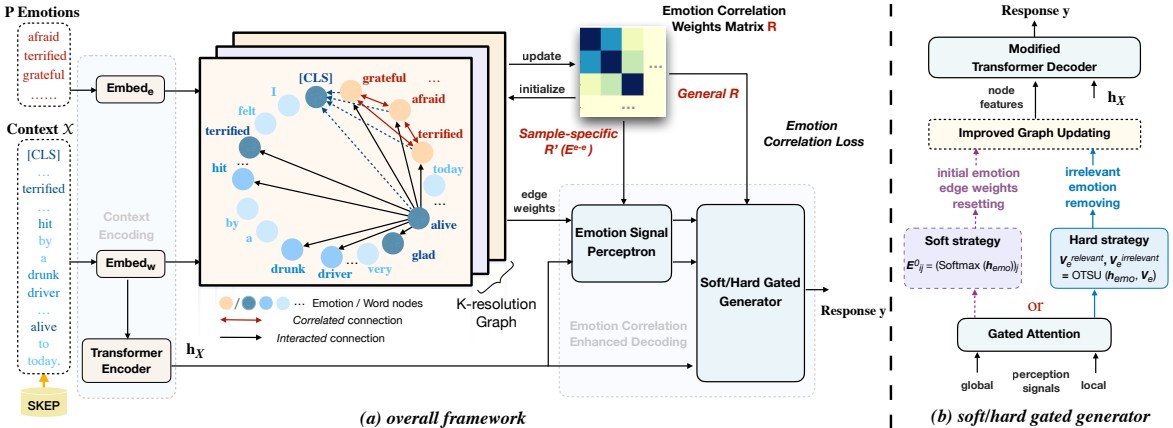

Figure 3: (a). The overview of the proposed E-CORE, which consists of three phases. 1) context encoding: encoding the dialogue context and all emotions into embedding features and contextual representation; 2) multi-resolution emotion graph network: capturing the context-based emotion interaction from different resolutions to encode the emotion correlation; 3) emotion correlation enhanced decoding: incorporating the emotion correlation to enhance emotion signal perception and response generation. (b). The design of soft/hard gated generator used in phase 3.

Our work more simulates real dialogue scenarios, imitating the listener's perception and inference for context emotions, which focuses on the multi-emotion learning with emotion correlation.

## 2.2 Empathetic Dialogue Generation

Unlike emotional dialogue generation, the empathetic dialogue generation task aims at generating empathetic responses, based on perceived emotions instead of definite annotated emotions. Rashkin et al. (2019) first proposed the task and contributed a new task benchmark and a large-scale empathy dialogue dataset. Then several works (Majumder et al., 2020; Li et al., 2020a; Kim et al., 2021; Gao et al., 2021) make efforts in enhancing empathy perception. Lin et al. (2019) proposed a multi-decoder model combining the emotional responses of the appropriate listeners, as every listener is independent. Kim et al. (2022) proposed a feature transition recognizer for identifying feature shifts between utterances, enhancing semantic understanding. Li et al. (2022b); Sabour et al. (2022) introduced commonsense knowledge to improve situation understanding. Li et al. (2022a) further proposed a serial encoding and emotion-knowledge interaction method which effectively utilized fine-grained emotion features and commonsense knowledge to enhance empathy response. However, these works mostly rely on single-emotion prediction to capture empathy signals, ignoring the emotion co-occurrence existing in dialogues. In this work, we investigate the correlation-based emotion co-occurrence to enhance empathetic perception and expression.

## 3 Proposed Approach

Given a dialogue context $\mathbf{U} = [u_1, u_2, \ldots, u_m]$ of $m$ utterances, empathetic dialogue generation aims to generate the next empathetic response $\mathbf{y}$ with emotional consistency and informative expression. Optionally, the task performs emotion prediction based on context semantic understanding to achieve empathetic constraints. In this section, we give a detailed introduction to our proposed E-CORE, which explicitly mines and incorporates the emotion correlation to enhance empathetic perception and expression. The framework consists of 3 phases: context encoding, multi-resolution emotion graph network, and emotion correlation enhanced decoding, as is illustrated in Fig.3.

## 3.1 Context Encoding

Following previous methods (Sabour et al., 2022; Li et al., 2022b), we first concatenate the dialogue context $\mathbf{U}$ into a long word sequence and insert a special $[CLS]$ token at the start, *i.e.,* $\mathbf{X} = [CLS, x_1, x_2, \ldots, x_{M-1}]$, where $M - 1$ is the total number of words in $\mathbf{U}$ and $x_0$ indicates $[CLS]$. Then we represent the context embedding as a synthesis of three kinds of embeddings: word embedding, position embedding (Vaswani et al., 2017) and dialog state embedding, as the dialog state indicates each word comes from the speaker or the listener. The context embedding $\mathbf{x}$ is fed into a transformer encoding layer (Vaswani et al., 2017)

to obtain the contextual representation:

$$\mathbf{x} = \mathbf{e_w}(\mathbf{X}) + \mathbf{e_p}(\mathbf{X}) + \mathbf{e_d}(\mathbf{X}), \qquad (1)$$

$$\mathbf{h}_X = \mathbf{Enc}_{trans}(\mathbf{x}). \qquad (2)$$

$\mathbf{h}_X \in \mathbb{R}^{M \times D}$ and $D$ is the feature dimension.

### 3.2 Multi-resolution Emotion Graph Network

Inspired by the methods in social psychology studies (Vansteelandt et al., 2005; Scherer, 2013) which explore the emotion correlated co-occurrence through emotion words interaction, we construct a multi-resolution emotion graph based on the word emotion intensities, to capture the context-based emotion interaction from different resolutions, for further emotion correlation learning.

Similar to Li et al. (2022b), we construct the emotion intensity annotation from SKEP (Tian et al., 2020), serving as the bridge for emotion graph modeling. As SKEP outputs a [0,1] score $\eta(x_i)$ identifying the positive degree of word $x_i$ (0.5 means neutral), the emotion intensity of each word is defined as $c_i = (\eta(x_i) - 0.5)^2$, and $\mathbf{c} = [c_1, \ldots, c_{M-1}]$ is the emotion intensity for all context words.

**Graph Construction.** Specially, the multi-resolution emotion graph is composed of two kinds of nodes, *i.e.*, $M$ *word nodes* $\mathbf{V}_w$ for $M$ context words (including $[CLS]$) and $P$ *emotion nodes* $\mathbf{V}_e$ for $P$ emotions; and two kinds of edges, *i.e.*, *interacted* connections for word nodes, and *correlated* connection for emotion nodes.

For word nodes, the emotion graph is required to capture the subtle emotions interaction existing in the context for correlation learning. Starting from the global interaction, as different emotional transitions will lead to different response emotions, we innovate a basic *interacted* connection, *i.e.*, a word node connects to previous word nodes and all emotion nodes. Further, to capture more direct emotion interactions, as the emotion intensity $\mathbf{c}$ preliminary indicates the word emotional importance, by setting different thresholds and screening out relatively unimportant word nodes, the basic graph will be extended to refined *interacted* graphs that attend to emotional information at multi-resolution.

For emotion nodes, the emotion graph is required to model the intrinsic emotion correlation, thus we construct the emotion *correlated* connection, *i.e*, edges from emotion nodes to each other, combined with a global learning matrix $\mathbf{R} \in \mathbb{R}^{P \times P}$, simply yet effectively encoding the correlation weights.

Considering the symmetry of emotion correlation, more generally, we adopt a re-parameterization trick to replace the direct training for $\mathbf{R}$, by representing $\mathbf{R}$ as the inner-product of the re-parameter matrix $\mathbf{S} \in \mathbb{R}^{P \times P}$, *i.e*, $\mathbf{R} = \mathbf{S}^\mathsf{T}\mathbf{S}$, where the diagonal values are always set to 1.

Logically, we define the initial edge weights for each node as the normalization for its corresponding neighboring nodes:

$$E_{ij}^0 = \begin{cases} c_j/\max(|\mathbf{c}|), & \text{for } v_i, v_j \in \mathbf{V}_w \\ 1/P, & \text{for } v_i \in \mathbf{V}_w, \ v_j \in \mathbf{V}_e \\ \text{Softmax}_j(\mathbf{R}), & \text{for } v_i, v_j \in \mathbf{V}_e \end{cases} \quad (3)$$

Additionally, all nodes are connected to $[CLS]$ node with weight 1 for context interaction. The initial features $\mathbf{h}^0$ for word nodes $\mathbf{V}_w$ and emotion nodes $\mathbf{V}_e$ are defined as the word embeddings $\mathbf{x}$ and emotion embeddings $\mathbf{e_w}(\mathbf{V}_e)$ (Eq.1).

**Graph Updating.** We design a novel multi-resolution attention mechanism that effectively realizes the independent updating and layer-out fusion of graph features for different resolutions, without increasing complexity. Specifically, the nodes and edges features of layer $l$ on $k$-th graph are updated:

$$\mathbf{h}_i^{l+1} = \mathbf{\Pi}^l[\mathop{\|}_{k=1}^{K} (\sum_{j \in \mathcal{N}_i^k} A_{ij}^{l,k} \mathbf{V}^{l,k} \mathbf{h}_j^l)], \quad (4)$$

$$E_{ij}^{l+1,k} = (\mathbf{W}_V^{l,k}\mathbf{h}_j^l) \odot (\mathbf{W}_E^l(E_{ij}^{l,k} + \hat{A}_{ij}^{l,k})), \quad (5)$$

$$\text{where, } A_{ij}^{l,k} = \text{Softmax}_{j \in \mathcal{N}_i^k}(\hat{A}_{ij}^{l,k}), \quad (6)$$

$$\hat{A}_{ij}^{l,k} = ((\mathbf{Q}^{l,k}\mathbf{h}_i^l)^\mathsf{T}\mathbf{K}^{l,k}\mathbf{h}_j^l) \odot E_{ij}^{l,k}, \quad (7)$$

where $K$ is the resolution level, $\mathbf{V}^{l,k}, \mathbf{Q}^{l,k}, \mathbf{K}^{l,k} \in \mathbb{R}^{(D/K) \times D}$, $\mathbf{W}_V^{l,k} \in \mathbb{R}^{1 \times D}$, $\mathbf{W}_E^l \in \mathbb{R}^{1 \times 1}$ are learnable matrixes. $A_{ij}^{l,k}$ indicates the calculated attention score of node $i$ to neighboring node $j$ on the $k$-th graph and $l$-th layer. $\odot$ denotes element-wise multiplication and $\mathbf{\Pi}^l$ is a MLP network. $\|$ denotes concatenation for each graph. This design smoothly promotes the multi-head attention into multi-resolution updating, where node features and edge weights are independently updated with corresponding connection in each resolution (head), then node features are fused layer-by-layer for global feature sharing.

After several rounds of graph updating and a sum process for $K$-graph edge weights, we obtain the representation of emotion graph: word-to-emotion edge weights $\mathbf{E^{w\text{-}e}} \in \mathbb{R}^{M \times P}$; emotion-to-emotion edge weights $\mathbf{E^{e\text{-}e}} \in \mathbb{R}^{P \times P}$ and word node features $\mathbf{h}_{node} \in \mathbb{R}^{M \times D}$, used for subsequent emotion perception and response generation.

## 3.3 Emotion Correlation Enhanced Decoding

With the sample-specific emotion correlation captured by graph, we detail the utilization of the correlation-based emotion co-occurrence, to enhance the emotion signal perception and empathetic response generation, respectively.

**Emotion Signal Perceptron.** We adopt correlation-aware aggregation to enhance emotion perception. Specifically, as the edge weights of graph intuitively reflect the attention to emotions, we define the global perception signal:

$$\mathbf{h}_{emo}^g = \mathbf{E}^{\mathbf{e\text{-}e}}(\sum_{i=1}^{M} \mathbf{E}_i^{\mathbf{w\text{-}e}}). \quad (8)$$

This processing refers to Fig.1-a, where the column-summation for $\mathbf{E}^{\mathbf{w\text{-}e}}$ fuses the attention weights of $M$ words to each emotion. $\mathbf{E}^{\mathbf{e\text{-}e}}$ is initialized by $\mathbf{R}$ and updated with sample context, equivalent to the sample-specific emotion correlation weights with diagonal values reset to 1. This design smoothly achieve an attention correlated aggregation for the co-occurrence emotions in the context.

Then the global perception signal is combined with contextual representation $\mathbf{h}_X$ (Eq.2), followed by a linear layer and a softmax layer to obtain the emotion category distribution. Specifically:

$$\mathbf{h}_{emo}^m = \mathbf{W}_\epsilon(\mathbf{h}_{emo}^g \,||\, \mathbf{W}_x \overline{\mathbf{h}}_X), \quad (9)$$
$$\mathcal{P}(\epsilon \mid \mathbf{X}) = \text{Softmax}(\mathbf{h}_{emo}^m), \quad (10)$$
$$\mathcal{L}_{emo} = -\log(\mathcal{P}(\epsilon = \epsilon^* \mid \mathbf{X}). \quad (11)$$

$\overline{\mathbf{h}}_X \in \mathbb{R}^D$ is the mean pooling feature of $\mathbf{h}_X$, $\mathbf{W}_\epsilon \in \mathbb{R}^{P \times 2P}$ and $\mathbf{W}_x \in \mathbb{R}^{P \times D}$ are weight matrixs of linear layers. $\mathbf{h}_{emo}^m$ is the obtained main perception signal. Our model minimize the cross-entropy loss between the predicted main-emotion $\epsilon$ and ground truth emotion $\epsilon^*$ for optimization.

**Soft/Hard Gated Generator.** Main-emotion signal perception provides annotated emotion supervision, but may also suppress other emotions, impairing subsequent generation. Thus, we design both soft and hard gated strategies to capture the meaningful co-occurrence emotions, combined with the emotion graph to pay more attention to meaningful emotions, and further achieve co-occurrence emotions guided generation.

Specifically, to avoid the supervised suppression may be caused by direct use of $\mathbf{h}_{emo}^m$, a gated attention mechanism is adopted to extract meaningful emotion features from the global and main emotion perception signals, which both contain rich emotional information:

$$\mathbf{h}_{emo} = \sigma(\mathbf{W}_e \mathbf{h}_{emo}^g) \odot \mathbf{h}_{emo}^m + \mathbf{h}_{emo}^m, \quad (12)$$

where $\mathbf{W}_e$ is weight metric and $\mathbf{h}_{emo} \in \mathbb{R}^P$ indicates the final attention features to $P$ emotions.

With the final emotion attention, soft and hard strategies are proposed respectively, to improve the graph for an effective utilization of correlated co-occurrence emotions. A straightforward way is *soft strategy*, which treats the attention features as an emotional soft label, serving as the new initial edge weight for emotion nodes:

$$E_{ij}^0 = (\text{Softmax}(\mathbf{h}_{emo}))_j, \text{ for } v_j \in \mathbf{V}_e. \quad (13)$$

However, the *soft strategy* may introduces redundant emotional information, resulting in noise interference. Therefore, we further propose another *hard strategy* to directly screen emotions. As the context-irrelevant/relevant emotions reflect a great distinction in attention features, we divide emotions into irrelevant and relevant categories, based on the principle of maximizing the variance between the two categories, also known as the OTSU algorithm (Otsu, 1979) (details in appendix B):

$$\mathbf{V}_e^{relevant}, \mathbf{V}_e^{irrelevant} = \text{OTSU}(\mathbf{h}_{emo}, \mathbf{V}_e). \quad (14)$$

By removing the nodes and connected edges of irrelevant emotions, *hard strategy* helps realize comprehensive attention to important emotions.

In summary, the *soft strategy* is more flexible while the *hard* is more stable, both of which successfully achieve an adaptive selection and utilization of co-occurrence emotions.

Finally, after carrying out soft or hard strategy to improve the emotion graph to focus on significant emotions, we obtain the improved graph features through another forward process, based on the parameters/weights-shared improved graph network. As node features reserve not only emotional information, but also emotion-interacted semantic information, we fed the improved node features $\hat{\mathbf{h}}_{node}$ into the modified transformer decoder (details in appendix C) for generation:

$$s_t = \mathbf{Dec}_{trans}^M(\mathbf{y}_{<\mathbf{t}}, \mathbf{h}_X, \hat{\mathbf{h}}_{node}), \quad (15)$$
$$\mathcal{P}(y_t \mid \mathbf{y}_{<\mathbf{t}}, \mathbf{X}) = \text{Softmax}(\mathbf{W}_s s_t), \quad (16)$$

where $\mathbf{y}_{<\mathbf{t}} = [y_0, ..., y_{t-1}]$ is the masked response and $\mathbf{h}_X$ is the contextual representation. As most dialogue generation tasks, the negative log-likelihood

loss is used as the optimization objective:

$$\mathcal{L}_{gen} = -\sum_{t=1}^{n} \log \mathcal{P}(y_t \mid \mathbf{y}_{<\mathbf{t}}, \mathbf{X}). \quad (17)$$

## 3.4 Emotion Correlation Loss

Finally, to avoid excessive or erroneous introduction of emotional information, we construct an emotion correlation loss for regular constraints:

$$\mathcal{L}_{eco} = -\frac{\sum_{v_i, v_j \in \mathbf{V}', i<j} \mathbf{R}[v_i, v_j]}{|\mathbf{V}'|} \quad (18)$$

where $\mathbf{V}'$ is the learned co-occurrence emotions, taking top-3 emotions for soft strategy and $\mathbf{V}_e^{relevant}$ for hard strategy. Obviously, minimizing $\mathcal{L}_{eco}$ loss prevent to introduce multi-emotion with low correlation weights, as low weights indicate the emotions are unlikely to occur in the same context.

Considering above all components, a joint loss function is adopted as the overall optimization objective to achieve end-to-end paradigm learning:

$$\mathcal{L} = \mathcal{L}_{gen} + \gamma_1 \mathcal{L}_{emo} + \gamma_2 \mathcal{L}_{eco} \quad (19)$$

## 4 Experiment Settings

### 4.1 Datasets

We evaluated our E-CORE on the EMPATHETIC-DIALOGUES (Rashkin et al., 2019) dataset, which is collected from Amazon Mechanical Turk and contains about 25k open-domain dyadic conversations. Each conversation comes from a speaker and a listener, in which the speaker is asked to talk about personal feelings, and the listener responds empathetically. We split the train/val/test set into $19,533/2,770/2,547$ conversations.

In addition, to further validate the fidelity of the E-CORE in emotion correlation modeling, we also construct a sub-dataset[2] with multi-emotion annotation. This sub-dataset is obtained by: 1) emotional annotating with large-scale language models *ChatGPT*(OpenAI, 2022) and *ChatLLaMa*(Nebuly-AI, 2023) on the above test set; 2) screening the samples that have identified the ground-truth emotion and contained multi-emotion labels; 3). filtering the mistaken annotation with manual inspection. This sub-dataset composes of 739 samples, with average of 2.93 emotion labels per sample.

[2]Sub-dataset detailed in appendix H.

## 4.2 Baselines

We conduct experiments to compare our E-CORE with the following state-of-the-art baselines: 1) **Transformer** (Vaswani et al., 2017): a transformer-based model for response generation. 2) **MIME** (Majumder et al., 2020): a model connsidering polarity-based emotion clusters and emotional mimicry. 3) **EmpDG** (Li et al., 2020a): a model exploiting multi-resolution emotions. 4) **KEMP** (Li et al., 2022b): a model introducing external knowledge. 5) **CEM** (Sabour et al., 2022): a model leveraging commonsense to draw more information. 6) **SEEK** (Li et al., 2022a): a model exploiting serial encoding and emotion-knowledge interaction. For a fair and clear comparison, without otherwise stated, all models and model variants of our E-CORE and SOTAs are trained from scratch based on dialogue-level emotion annotations.

Our model is explored using soft and hard strategies respectively, as introduced in Sec.3.3, denoted as Ours(Soft) and Ours(Hard). The model is based on the transformer (Vaswani et al., 2017) framework with 4 blocks and 3 heads, with the emotion graph of layer $L = 2$ and resolution level $K = 3$, corresponding to the threshold $[0, 0.075, 0.15]$. The parameters for loss function are $\gamma_1 = \gamma_2 = 1$. More implementation details are covered in appendix D.

### 4.3 Evaluation Metrics

**Automatic Evaluation.** Following previous works, for response generation, we adopt perplexity (**PPL**) (Serban et al., 2015) and distinct-n (**Dist-n**) (Li et al., 2015) as the main automatic metrics which measures the quality and diversity of generated responses, respectively. For emotion perception, we employ the emotion accuracy (**Acc**) to measure the consistency between predicted main-emotion and ground-truth emotion.

**Human Evaluation.** To test the model's ability on generating human-like responses, we conduct **human ratings** to evaluate the generated responses from three aspects: **Fluency** (fluency of responses), **Relevance** (relevant to dialogue context) and **Empathy** (empathetic expression of responses). We randomly select 100 dialogues, paired with the dialogue context and responses from the baselines and our E-CORE. Three human annotators are asked to score the selected instances on three metrics in the range of $[1, 5]$, with the higher the better. The average scores of all annotators are the human rating

| Models | Automatic Evaluation | | | | Human Evaluation | | |
|---|---|---|---|---|---|---|---|
| | PPL↓ | Dist-1 | Dist-2 | Acc | Fluency | Relevance | Empathy |
| Transformer (Vaswani et al., 2017) | 37.73 | 0.47 | 2.04 | – | 3.76 | 3.32 | 3.14 |
| MIME (Majumder et al., 2020) | 37.09 | 0.47 | 1.91 | 34.24 | 3.82 | 3.64 | 3.35 |
| EmpDG (Li et al., 2020a) | 37.29 | 0.46 | 2.02 | 34.31 | 3.79 | 3.67 | 3.46 |
| SEEK (Li et al., 2022a) | 37.37 | 0.70 | 3.13 | 38.90 | 3.86 | 3.78 | 3.54 |
| KEMP (Li et al., 2022b) | 36.89 | 0.55 | 2.29 | 39.31 | 3.88 | 3.86 | 3.62 |
| CEM (Sabour et al., 2022) | 36.11 | 0.66 | 2.99 | 39.11 | 3.92 | 3.77 | 3.60 |
| Ours(Soft) | 33.04 | 0.68 | 3.38 | 42.57 | **3.94** | 3.96 | 4.02 |
| Ours(Hard) | **33.03** | **0.72** | **3.49** | **42.59** | 3.92 | **4.00** | **4.08** |

Table 1: Comparisons with SOTAs. ↓ suggests that the performance is better with a lower score.

| Models | Win % | Loss % | Tie % |
|---|---|---|---|
| Ours(Soft) vs KEMP | 38.6 | 19.1 | 42.3 |
| Ours(Soft) vs CEM | 37.9 | 19.4 | 42.7 |
| Ours(Hard) vs KEMP | 42.6 | 19.5 | 37.9 |
| Ours(Hard) vs CEM | 42.9 | 19.7 | 37.4 |
| Ours(Soft) vs Ours(Hard) | 26.7 | 32.1 | 41.2 |

Table 2: Comparisons with SOTAs on human A/B test.

| Models | PPL↓ | Dist-1 | Dist-2 | Acc | R@1 | R@3 | R@5 |
|---|---|---|---|---|---|---|---|
| MIME | 35.18 | 1.25 | 3.27 | 33.05 | 0.51 | 0.79 | 1.03 |
| EmpDG | 35.27 | 1.23 | 3.20 | 33.17 | 0.52 | 0.84 | 1.07 |
| SEEK | 35.14 | 1.78 | 6.54 | 34.22 | 0.51 | 0.88 | 1.12 |
| KEMP | 34.56 | 1.39 | 4.27 | 36.37 | 0.58 | 1.02 | 1.32 |
| CEM | 34.52 | 1.64 | 5.37 | 36.26 | 0.57 | 0.98 | 1.24 |
| Ours(Soft) | **31.12** | **2.36** | **9.57** | 39.41 | **0.65** | **1.47** | **1.92** |
| Ours(Hard) | 31.18 | 2.11 | 8.62 | **40.46** | — | 1.47 | — |

Table 3: Comparisons with SOTAs on the sub-dataset.

results. In addition, for more direct model comparison, we also conduct the **human A/B test** with the best-performing SOTAs. Three annotators are required to carry out pairwise response comparisons, selecting better response for each instance. Tie is allowed if both are good or bad. More details for human evaluations are covered in appendix I.

## 5 Results and Analysis

We conduct experiments on the benchmark dataset to verify the promise of emotion correlation learning in both emotion perception and empathetic generation. Then we investigate the ability of co-occurrence emotions recognition on the multi-emotion annotated subset, to further validate the essence of emotion correlation learning in our method E-CORE.

### 5.1 Comparison with State-of-the-Art

**Automatic Evaluation.** As SOTAs are mainly trained from scratch, we report the results trained from scratch for comparison fairness in Tab.1.[3] Our proposed E-CORE exhibits better performances than SOTAs on all automatic metrics, verifying the effectiveness of emotion correlation modeling in empathetic understanding. The significant improvements in response quality (**8.53%** in relative, **3.08** in absolute for **PPL**) and diversity (**11.5%** in relative, **0.36** in absolute for **Dist-2**) show that

---
[3]Results on pre-trained model are supplied in appendix E.

E-CORE generates more relevant comments rich in diversity, as more sufficient emotional information is provided. The great promotion of emotion accuracy (**8.34%** in relative, **3.28** in absolute for **Acc**) proves that, adopting correlation-aware aggregation rather than simply separating emotions, is beneficial for main-emotion perception.

**Human Evaluation.** The right part of Tab.1 presents the human rating results. E-CORE achieves better performance in all aspects, especially in **relevance** and **empathy**. The great improvement in **relevance** and **empathy** indicates that our model with emotion correlation learning helps provide more relevant emotions guidance, yielding more humanized and empathetic responses rich in relevant emotions and semantics. In addition, the pairwise comparison results in Tab.2 also confirm that the responses generated by E-CORE are preferred by human judges, either using soft strategy or hard.

**Results on Sub-dataset.** The current evaluation confirms that, even for EmpDG under single-label guidance, our model effectively achieves co-occurrence emotions learning with correlation modeling. To further validate the ability of our multi-emotion learning, we conduct extensive experiments on the multi-emotion annotated sub-dataset, using the metric **Recall@k** for quantitative evaluation, which indicates the number of ground-truth emotions covered by the top-k predicted emotions

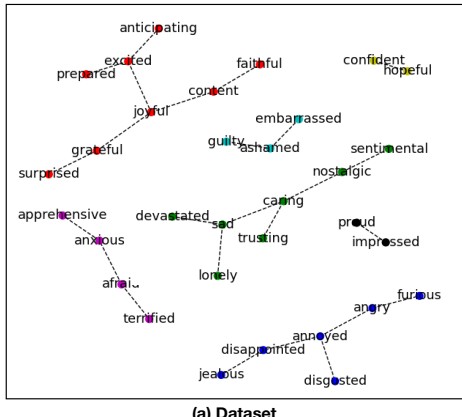
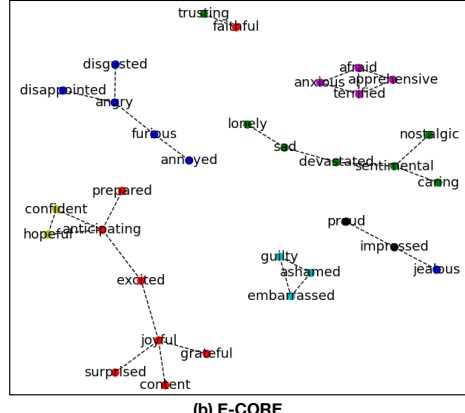

(a) Dataset

(b) E-CORE

Figure 4: Visualizations of emotion correlation for dataset and E-CORE. Displayed edges are between emotions with correlation weights greater than 0.3 after maximum-value normalization. Same emotions in (a) & (b) are highlighted in the same color, which is marked based on the emotion distribution in the dataset.

| Models | PPL↓ | Dist-1 | Dist-2 | Acc |
|---|---|---|---|---|
| Ours(Soft) | **33.04** | **0.68** | **3.38** | **42.57** |
| w/o graph | 36.42 | 0.51 | 2.30 | 40.36 |
| w/o co-p | 35.92 | 0.53 | 2.38 | 39.26 |
| w/o co-loss | 35.36 | 0.56 | 2.71 | 38.15 |
| Ours(Hard) | **33.03** | **0.72** | **3.49** | **42.59** |
| w/o graph | 36.44 | 0.52 | 2.98 | 39.83 |
| w/o co-p | 35.71 | 0.51 | 2.55 | 39.92 |
| w/o co-loss | 35.50 | 0.50 | 2.36 | 38.19 |
| w/o co-g | 35.45 | 0.54 | 2.70 | 42.00 |

Table 4: Results on ablation studies.

| Emotion | Sentimental |
|---|---|
| Context | **Speaker$_1$**: I went through some of my old stuff yesterday, and I found my security blanket that I used when I was a kid! |
| MIME | I am sure you will do great. |
| EmpDG | That is so sweet. |
| SEEK | That is so great. |
| KEMP | I am glad you are able to get it fixed. |
| CEM | I am sure you will get a good time. |
| **Ours**(soft) **Ours**(hard) | That is so nice of you to go back memories. I also love those moments. (Relevant Emotion: Sentimental, Nostalgic) |
| Gold | **Awww I bet that brought back memories.** |

Table 5: Case study of the generated responses by our E-CORE and the baselines.

(or the predicted relevant emotions for hard strategy). The great promotion shown in Tab.3 (**44.1%** in **R@3**, **31.3%** in **R@5**) reflects the significant superiority of E-CORE on multi-emotion learning. In addition, a greater improvement in original metrics (**11.6%** in **Acc**), further proves that our E-CORE has a stronger learning ability for complex samples with multiple emotions over SOTAs.

Further, we visualize the emotion correlation of the dataset and E-CORE for a more intuitive comparison. Among these, the correlation weight for the dataset is calculated based on the co-occurrence counts of emotion pairs, and for E-CORE is directly using the learned weight **R** after model training. As shown in Fig.4, our model shows a very close emotion correlation to real distribution, proving the accuracy of emotion correlation modeling.

## 5.2 Ablation Study

To fully examine the contribution of each design in Our E-CORE for addressing corresponding challenges, we conduct ablation studies through the

following variants: 1) **w/o graph**: the model without multi-resolution emotion **graph**, which directly implements other modules with the vanilla transformer framework. 2) **w/o co-p**: the model without **co**rrelation-aware aggregation in emotion **p**erceptron. 3) **w/o co-g**: the model without **co**rrelated co-occurrence emotions guidance in **g**enerator, which not uses soft/hard strategy and generates responses with main-emotion. 4) **w/o co-loss**: the model without emotion **co**rrelation **loss**.

As reported in Tab.4, all modules make reasonable contributions to E-CORE. For **learning**, replacing the emotion graph with transformer causes significant performance degradation, verifying the effectiveness of multi-resolution emotion graph for emotion correlation learning. For **utilizing**, models without correlation utilizing on perceptron or generator respectively perform weakly in emotion accuracy and response quality, indicating that our designed aggregation and soft/hard strategies effectively incorporate the correlated co-occurrence

emotions to enhance empathetic perception and expression. Finally, the result without correlation loss proves its importance for global **supervision**. More ablation studies and analyses in appendix J.

### 5.3 Case Study

As the case in Fig.1 has shown the ability of E-CORE to jointly guide generation with captured very different co-occurrence emotions (*afraid* and *grateful*), Tab.5 exhibits a case with similar co-occurrence emotions for comprehensive qualitative analysis. As the speaker expresses *sentimental* for "old stuff", relying on the significant correlation between *sentimental* and *nostalgic*, our E-CORE successfully identifies the auxiliary emotion *nostalgic*, generating more relevant phrases "go back memories" and "those moments", while the baseline models only produce universal responses. In general, whether similar or distant co-occurrence emotions are significant for EmpDG, which all help for global and detailed empathetic expression. Our E-CORE with emotion correlation learning helps provide sufficient emotion guidance, yielding more humanized responses rich in empathy.

## 6 Conclusion

In this paper, we propose to exploit the *intrinsic emotion correlation* in dialogues to enhance empathetic dialogue generation. A distinctive framework with three effective modules respectively addressing the emotion correlation learning, utilizing, and supervising, is designed. Extensive experiments on the benchmark dataset prove the significant advantages of our framework in improving emotion perception and empathetic generation. Specific analysis further demonstrates the accuracy of our emotion correlation learning. In the future, our work can inspire other approaches to explore emotion-related tasks with multi-emotion correlation learning, without being limited by single-emotion label.

## Limitations

1) Firstly, as we analyzed in the introduction, almost all dialogues are accompanied by subtle emotions besides the main-emotion. However, it is almost impossible to annotate all subtle emotions and even the emotion weights for a dialogue. Although our method based on emotion correlation modeling has effectively achieved multi-emotion learning for EmpDG under single-label guidance, how to improve the network to utilize existing in-

formation to provide more effective supervision for multi-emotion learning still needs to be considered. This is also a common problem faced by many emotion-related generation tasks. The ablation study on the model without response reconstruction loss supervision shown in appendix J indicates that the supervision for multi-emotion partly sources from the empathetic response, which may serve as an improved inspiration. 2) Secondly, all existing methods are evaluated on the unique benchmark dataset EMPATHETICDIALOGUES (Rashkin et al., 2019). As empathetic dialogue generation is an emerging task, only one relevant English dataset has been proposed, lacking of datasets in more languages and categories for reference. 3) Finally, we observed in the experiment that the existing models tend to generate generic responses, especially for complex hard samples, which are difficult to capture the key points. Therefore, the learning of hard samples is also a developing direction of the empathetic dialogue generation task.

## Ethics Considerations

The empathetic dialogue dataset (Rashkin et al., 2019) used in our paper is publicly-available and annotated through Amazon Mechanical Turk, which means it totally protects the privacy of real users. Besides, we make sure the anonymization in the human evaluation process. We believe our research work meets the ethics of EMNLP.

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

## A Statistics for Dialogue Emotions

To verify the importance of the multi-emotion correlation for the empathetic dialogue generation task, we make statistics on the quantity of emotion-related words of other emotions contained in the dialogue samples of the benchmark dataset EMPATHETICDIALOGUES (Rashkin et al., 2019), to preliminary observe the emotions co-occurrence situation in the dataset. Specifically, our annotators provide the annotation of high-frequency emotion-related words for all 32 emotions in the dataset. By counting the number and frequency of emotion-related words of different emotions in the dialogue, we can roughly inform the co-occurrence situation of various emotions in the dataset. The annotated emotion-related words of 32 emotions are shown in Tab.12.

## B Details of Hard Strategy

In this section, we elaborate on the hard strategy, which adopts the OTSU algorithm, *i.e.*, maximizing the variance between the two categories to divide emotions into irrelevant and relevant categories:

$$\mathbf{V}_e^{relevant}, \mathbf{V}_e^{irrelevant} = \text{OTSU}(\mathbf{h}_{emo}, \mathbf{V}_e). \quad (20)$$

Taking final emotion attention $\mathbf{h}_{emo}$ as the emotion attention value for emotion nodes $\mathbf{V}_e$, specifically, the above equation can be written in detail:

$$\mathbf{V}_e^{relevant} = \arg \max_{\mathbf{V}} \left( \frac{|\mathbf{V}|}{|\mathbf{V}_e|} ||\mu - \mu^{\mathbf{V}}||^2 \quad (21) \right.$$

$$\left. + \frac{|\overline{\mathbf{V}}|}{|\mathbf{V}_e|} ||\mu - \mu^{\overline{\mathbf{V}}}||^2 \right), \; \overline{\mathbf{V}} = \complement_{\mathbf{V}_e} \mathbf{V}.$$

Where $\mu, \mu^{\mathbf{V}}, \mu^{\overline{\mathbf{V}}}$ is the mean attention values of emotion nodes in the corresponding set $\mathbf{V}_e, \mathbf{V}, \overline{\mathbf{V}}$. In the formula, there is no obvious distinction between $\mathbf{V}$ and $\overline{\mathbf{V}}$, and we set the part with larger attention values corresponding to $\mathbf{V}$.

Obviously, for $N$ emotions, there are at most $N$ segmentation thresholds, as the emotions with a higher attention feature than the threshold will be regarded as the relevant category. Based on the statistical results shown in Fig.2, there are almost no five or more emotions in the dialogues in the dataset. To facilitate the operation, only the first five segmentation thresholds are considered, comparing their inter-categories variance to obtain the optimal division of emotion.

## C Modified Transformer Decoder

In this section, we provide a detailed introduction for the modified transformer decoder used in the soft/hard gated generator:

$$s_t = \mathbf{Dec}_{trans}^M(\mathbf{y}_{<t}, \mathbf{h}_X, \hat{\mathbf{h}}_{node}). \quad (22)$$

Taking masked response $\mathbf{y}_{<t} = [y_0, ..., y_{t-1}]$, contextual representation $\mathbf{h}_X$, improved graph node feature $\hat{\mathbf{h}}_{node}$ as the input, the detailed implementation is:

$$\mathbf{h}_X^t = \text{MH-ATT}(\mathbf{y}_{<t}, \mathbf{h}_X), \quad (23)$$

$$\hat{\hat{\mathbf{h}}}_{node} = \sum_{i=1}^M (\hat{\mathbf{h}}_{node})_i, \quad (24)$$

$$\hat{s}_t = \text{LayerNorm}(\mathbf{y}_{<t} + \mathbf{W}_d(\mathbf{h}_X^t || \hat{\hat{\mathbf{h}}}_{node}), \quad (25)$$

$$s_t = \text{LayerNorm}(\hat{s}_t + \text{FFN}(\hat{s}_t)), \quad (26)$$

where MH-ATT and FFN denote multi-head attention layer and feed-forward network respectively. Through a simple concatenation operation, the modified transformer decoder effectively introduce the graph node feature which is rich in emotion-interacted semantic information into the decoding process.

## D Implementation

The model is implemented in PyTorch (Paszke et al., 2017) with a single NVIDIA GeForce RTX 3090 GPU, and trained for about 10 epochs with batch size 16 and dropout rate 0.2. The training time of E-CORE is about 3 hours for around 26000 iterations. The vocabulary size is $23, 714$, and use the pre-trained Glove vectors (Pennington et al., 2014) for word embedding initialization. Our model is optimized by Adam optimizer (Kingma and Ba, 2014) and the learning rate is changed during training according to Vaswani et al. (2017) with the final learning rate is 3.5e-4. We also introduce the commonsense knowledge and the label smoothing strategy used in the SOTA model (Li et al., 2022b) as a trick to improve performance, without losing comparative fairness.

## E Experiments based on Pre-trained Model

As SOTAs are all trained from scratch, for comparison fairness, we mainly report the results model trained from scratch in the main body. In this section, we further explore our E-CORE and

| Models | PPL↓ | Dist-1 | Dist-2 | Acc |
|---|---|---|---|---|
| KEMP-DialoGPT | 15.21 | 2.79 | 4.24 | 46.43 |
| CEM-DialoGPT | 15.06 | 2.88 | 4.52 | 46.20 |
| Ours(Soft)-DialoGPT | 13.02 | 2.96 | 4.91 | 50.04 |
| Ours(Hard)-DialoGPT | **12.94** | **3.07** | **4.96** | **50.12** |

Table 6: Comparisons results based on pre-trained language model.

| Models | PPL↓ | Dist-1 | Dist-2 | Acc |
|---|---|---|---|---|
| Ours(Soft) | 33.04±0.22 | 0.68±0.01 | 3.38±0.02 | 42.57±0.13 |
| Ours(Hard) | 33.03±0.24 | 0.72±0.02 | 3.49±0.04 | 42.59±0.09 |

Table 7: Results of variance on all evaluation metrics.

SOTAs using a pre-trained language model DialoGPT (Zhang et al., 2020b). As shown in Tab.6, all models have a significant improvement while using the pre-trained language model, indicating that the huge amount of pre-trained dialogue datasets is beneficial for the empathetic dialogue generation task. Besides, our E-CORE consistently shows superior performance over SOTAs, which demonstrates the advantages of our model over SOTAs whether uses a pre-trained model or not.

## F  Experiments on Stability Testing

To verify the stability of the model, we evaluate the variance and statistical significance for E-CORE. Specifically, we adopt 5 different random seeds to conduct experiments on our model, based on soft and hard strategies respectively. Tab.7 reports the variance of all evaluation metrics, verifying the performance stability of the model.

## G  Statistical Significance

For statistical significance, we conduct one-side Student's t-test, proving our model (soft) significantly outperforms the best-performing baseline CEM with $p = 5.68$e-5 ($p < 0.05$ indicates that the hypothesis that A (E-CORE) outperforms B (CEM) is significantly valid). These conclusions hold for hard strategy.

## H  Sub-dataset with Multi-emotion Annotation

We construct a sub-dataset with multi-emotion annotations as an auxiliary test set of the benchmark dataset EMPATHETICDIALOGUES, to verify the accuracy of emotion correlation modeling in our E-CORE. In this section, we will provide a detailed explanation for this sub-dataset.

Firstly, based on the large-scale pre-trained language models *ChatGPT* (OpenAI, 2022) and *ChatLLaMa* (Nebuly-AI, 2023), we conduct emotion annotation for all $2,547$ conversations of the test set of EMPATHETICDIALOGUES, obtaining an intermediate dataset of $1,536$ samples labeled with multiple emotions. Secondly, we screen a total of $1,097$ samples that successfully identify the ground-truth emotion, which are proven to have higher annotation quality. Finally, a manual examination is conducted to filter out mistaken annotations. The final dataset is composed of 739 samples, with an average of 2.93 emotion labels per sample. Tab.11 shows some dialogue samples of this auxiliary dataset.

## I  Details of Human Evaluation

To evaluate the model's ability on generating human-like responses, we conduct experiments on human evaluation from three aspects: **Fluency**, **Relevance** and **Empathy**. Three human annotators are asked to score the instances on these there aspects in the range of [1,5]. We use Spearman's Rank correlation coefficients to evaluate the agreement among the annotators. The coefficients between any two annotators are all near 0.6 and at an average of 0.64, which shows the consistency and reliability of human evaluation scores.

In the following, we further provide the guidelines regarding how to judge the quality of the model's result on these three aspects in terms of different features.

### I.1  Fluency

This metric measures the fluency of the model's result. The definitions of different scores are:

- **[5]**: The generated responses are human-like, grammatically correct, fluent, and very easy to understand.

- **[4]**: Choose this score when you are hesitant between the score 3 and score 5.

- **[3]**: The generated responses have a few grammar errors, but not hinder understanding.

- **[2]**: Choose this score when you are hesitant between the score 1 and score 3.

- **[1]**: The generated responses have numerous grammar errors and difficult to understand.

| Models | PPL$^\downarrow$ | Dist-1 | Dist-2 | Acc |
|---|---|---|---|---|
| Ours(Soft) | **33.04** | **0.68** | **3.38** | **42.57** |
| w/o gate-g | 35.11 | 0.58 | 3.01 | 42.02 |
| w/o gen-loss | 34.40 | 0.61 | 2.73 | 39.37 |
| Ours(Hard) | **33.03** | **0.72** | **3.49** | **42.59** |
| w/o gate-g | 35.32 | 0.56 | 3.07 | 42.13 |
| w/o gen-loss | 34.74 | 0.57 | 2.50 | 39.83 |

Table 8: More results on ablation study.

## I.2 Relevance

This metric measures the informativeness and relevance of the model's result. The definitions of different scores are:

- **[5]**: The generated responses are perfectly related to the dialogue context.

- **[4]**: Choose this score when you are hesitant between the score 3 and score 5.

- **[3]**: The generated responses are to some extent related to the dialogue context.

- **[2]**: Choose this score when you are hesitant between the score 1 and score 3.

- **[1]**: The generated responses are completely unrelated to the dialogue context.

## I.3 Empathy

This metric measures the empathy of the model's result. The definitions of different scores are:

- **[5]**: The generated responses are rich in emotional expression, and the expressed emotions perfectly correspond to the dialogue context.

- **[4]**: Choose this score when you are hesitant between the score 3 and score 5.

- **[3]**: The generated responses to some extent contain the emotional expression, and the expressed emotions to some extent correspond to the dialogue context.

- **[2]**: Choose this score when you are hesitant between the score 1 and score 3.

- **[1]**: The generated responses do not contain the emotional expression, or the expressed emotions do not correspond to the dialogue context.

| Models | PPL$^\downarrow$ | Dist-1 | Dist-2 | Acc |
|---|---|---|---|---|
| Ours(Soft)-SKEP | **33.04** | **0.68** | **3.38** | **42.57** |
| VAD | 33.39 | 0.66 | 3.36 | 42.28 |
| VADER | 33.35 | 0.65 | 3.32 | 42.17 |
| SentiWordNet | 33.46 | 0.63 | 3.34 | 42.05 |
| w/o | 34.97 | 0.54 | 3.02 | 40.06 |
| Ours(Hard)-SKEP | **33.03** | **0.72** | **3.49** | **42.59** |
| VAD | 33.35 | 0.68 | 3.44 | 42.35 |
| VADER | 33.32 | 0.66 | 3.42 | 42.07 |
| SentiWordNet | 33.42 | 0.65 | 3.38 | 42.02 |
| w/o | 34.91 | 0.57 | 3.06 | 40.12 |

Table 9: More ablation studies on the performance of emotion intensity, where SKEP is used for original method.

## J  More Ablation Study

To further examine our E-CORE, more ablation studies are conducted through following variants: 1) **w/o gate-g**: for testing the sub-module of soft/hard gated **g**enerator, the model without **gate**d attention, directly using the main emotion feature for guidance. 2) **w/o gen-loss**: for testing the sources of emotional supervision, the model without response generation loss $\mathcal{L}_{gen}$.

As we can see in Tab.8, all sub-modules contribute a lot to the whole model. The gated attention mechanism has a great impact on generation, suggesting that gated attention is helpful for meaningful emotions extracting, which further provide more sufficient emotional guidance to enhance expression. It is worth noting that models without response generation loss (**w/o gen-loss**) not only show a decline in the generation, but also perform poorly in the emotion prediction (**42.57** to **39.37** in **Acc**), indicating that emotion supervision not only comes from the single-emotion label, but also from the empathetic responses, further proving the reliability of our emotion correlation learning, which modeled by emotion graph with context-based interaction capturing and multi-sample joint transfer learning.

## J.1  More Ablation Study on Emotion Intensity

We also conduct additional ablation studies to evaluate the performance of emotion intensity, by comparing the variants without emotion intensity or with different emotion intensity labeling on both soft and hard strategies. The different emotion intensity values are obtained by four different emotion analysis models, specifically: SentiWordNet

| Emotion | Proud |
|---|---|
| Context | **S₁**: Lately I have felt proud of my success as a newly single mother. It gets lonely sometimes, but I can honestly say I have been doing everything I can and more. |
| Gold | **Keep up the good work! You will do good in life.** |
| MIME | I am so happy for you. |
| SEEK | That is grea for you. |
| EmpDG | That is great! I hope you get it! |
| KEMP | That is great to hear! I hope you can find a new job. |
| CEM | That is good to hear. |
| **Ours**(soft) | I am sure you are a great parent! |
| **Ours**(hard) | I am sure you will be proud of yourself. |
| | (Relevant Emotion: **Proud**) |
| Emotion | Faithful |
| Context | **S₁**: My husband is the most faithful man. |
| | **L₁**: That is great to hear! A faithful spouse is a blessing. |
| | **S₂**: I have so many health problems and he is always there for many no matter what being loving and caring. |
| Gold | **I am sorry to hear about that! I hope everything gets better for you!** |
| MIME | Oh no! That is terrified for you. |
| EmpDG | That is great for you! |
| SEEK | I hope you can get better. |
| KEMP | I agree with you. I am sure he will have a great relationship. |
| CEM | That is good. I hope he gets better. |
| **Ours**(Soft) | That is great! Hope you can get through that! |
| **Ours**(Hard) | That is great! I hope you have a great life! |
| | (Relevant Emotion: **Faithful**, **Hopeful**) |

Table 10: More case studies of the generated responses by our E-CORE and the baselines. Key words in context and responses of different emotions are highlighted in different colors. $S_i$ and $L_i$ respectively correspond to the $i$-th sentence from the speaker or listener.

(Sebastiani and Esuli, 2006), VADER (Hutto and Gilbert, 2014), VAD (Zhong et al., 2019), SKEP (Tian et al., 2020).

As we can see from Tab.9, the model without emotion intensity, which degenerates into a single-resolution emotion graph, performs weakly, indicating the great impact of multi-resolution graph modeling for correlation learning. Furthermore, the model exhibits strong robustness to different emotion intensity labeling, also indicating that the emotion correlation learning more relies on graph training, instead of the performance of the original emotion intensity.

## K  More Case Studies

Typically, two more generation cases of single-emotion guidance and double-emotion guidance are shown in Tab.10. In the first case, E-CORE extracts the key information "proud" and "single mother" from the context, and generates more detailed and accurate phrases "great parent" and "proud", while the baseline models only predict a generic phrase "great". In the second case, the speaker expresses her husband's "faithful" and her "health problems". Our E-CORE successfully detects two emotions in the dialogue context, generating the praise "great" for *faithful* and wishes "get

through" or "great life" for *hopeful*, while other baselines either take a wrong understanding or only notice one emotion. In general, the emotion correlationn learning enhances the emotion-interacted semantic understanding, resulting in more humanized responses rich in information and empathy, whether in simple or complex dialogue contexts.

## L  Visualization Analysis

To further explore the working mechanism of our emotion graph, we visualize the edge features from dialogue word nodes to 32 emotion nodes for the **case** of Fig.1. As shown in Fig.5, our Ours(Soft) puts the highest attention on the words containing informative meaning, among which the words "terrified", "hit" and "drunk driver", contribute to emotion *terrified* and *afraid*, as the words "glad" and "alive" pay more attention to *grateful*. We can conclude that our multi-resolution emotion graph effectively learns diverse emotional information.

In addition, we also explored the effect of correlation-aware aggregation for emotion perception, and visualize the working mechanism in Equation 8 based on the same case as above. As we can see from Fig.6, for the initial graph word-emotion edge fusion features, similar emotion *terrified* was mistakenly selected while the ground-truth emotion

is *afraid*. Although *afraid* is already very close, for the existing methods, it will be discarded while it is recognized as a secondary emotion. However, in our E-CORE, after carrying out correlation-aware aggregation, the greater weight of *grateful* for *afraid* assists in identifying the *afraid* as the main-emotion, achieving full utilization of all co-occurrence emotions. This further confirms the effectiveness of correlation-based emotion co-occurrence learning in enhancing emotional perception, especially for hard samples with similar emotions.

| Emotion | **Jealous**, **Grateful**, **Hopeful** |
|---|---|
| **Context** | $S_1$: My sister recently paid off her house. I felt so envious of her as i have been trying to pay mine down.
$L_1$: That will be such a wonderful feeling when you do. I know it is hard when you see someone reach the finish line first. Are you able to turn that envy into inspiration?
$S_2$: I really am trying hard to. That is such a great suggestion. I know i am making good progress to my goal, and I genuinely am pleased she did it!
$L_2$: See? That is terrific! And she 's got to love that you can celebrate her victories too – even more inspiration to support yours! |
| **Emotion** | **Disappointed**, **Content**, **Joyful** |
| **Context** | $S_1$: We went on our vacation about a week ago to the beach and it rained several days while we were there. We wanted to go to the beach every day!
$L_1$: I hate when that happens! Especially when you have been waiting for the beach!
$S_2$: Yes, and when you pay a fortune to get beachfront! But , we still had a good time, just wished we could have had less rain!
$L_2$: That is what matters most, that you had a good time and made memories! |
| **Emotion** | **Sentimental**, **Nostalgic** |
| **Context** | $S_1$: I was going through some boxes the other day. I found some old pictures of my kids I thought were gone.
$L_1$: That is exciting! I love having pictures to look back on.
$S_2$: Yes, definitely! It really made us all start talking about those and other memories, good times.
$L_2$: I love being able to reminisce on the past. Time goes by so fast. |
| **Emotion** | **Hopeful**, **Anticipating**, **Proud** |
| **Context** | $S_1$: I have been extremely diligent in my studies so far, and am participating in two clubs this upcoming year. The future is looking bright!
$L_1$: That is awesome. What are you hoping to do?
$S_2$: I am an electrical engineering student, and i am interested in quite a few things at this point. I think after this year, I will be able to get a paid internship next summer!
$L_2$: That is the dream, to get paid in college. I wish I could have done that. Congratulations! |
| **Emotion** | **Confident**, **Proud** ,**Impressed** |
| **Context** | $S_1$: My daughter is such a talented and creative young artist. When she was nominated to be 'most artistic' at school, I felt very secure she would win, because she really is amazing!
$L_1$: Thats awesome. Did she?
$S_2$: She did! I know I am the parent so can be biased, but it is really so impressive how well she draws, paints, shades . . . .
$L_2$: Bet you are so proud. I would definitely be. |

Table 11: **Sub-dataset.** Example dialogues in our constructed multi-emotion annotated subset. The ground-truth emotion is highlighted in red. $S_i$ and $L_i$ respectively correspond to the $i$-th sentence from the speaker or listener.

| surprised | surprised, astonish, astound, amaze, startle |
|---|---|
| excited | excited, aroused, ecstatic, elated, rapturous, euphoric, exhilarated |
| annoyed | annoyed, irritated, miffed, nettled, peeved, pissed, riled, roiled, stung |
| proud | proud, gallant, lofty, majestic, arrogant, haughty |
| angry | angry, mad, indignant, cross, irate |
| sad | sad, deplorable, distressing, lamentable, pitiful, sorry |
| grateful | grateful, gratitude, grate, thankful, appreciative |
| lonely | lonely, alone, lone, solitary, unfrequented, lonesome, desolate |
| impressed | impressed, awed, overcome, overwhelmed, dazzled, stunned |
| afraid | afraid, scared, alarmed, petrified, fearful |
| disgusted | disgusted, sick of, tired of, outraged, appalled, offended, sickened, scandalized |
| confident | confident, sure, convinced, certain, positive |
| terrified | terrified, panicky, panic-stricken, fright, frighted |
| hopeful | hopeful, optimistic, promising, hope, aspirant, bright |
| anxious | anxious, concerned, nervous, uneasy, worried |
| disappointed | disappointed, defeated, discomfited, foiled, frustrated, thwarted |
| joyful | joyful, pleasant, agreeable, cheerful, joyful, elated, gleeful, jubilant |
| prepared | prepared, ready, disposed, inclined |
| guilty | guilty, criminal, blameworthy, fault, culpable, hangdog, shamefaced, shamed |
| furious | furious, enraged, ferocious, fierce, savage, infuriated, maddened, raging, tempestuous, wild |
| nostalgic | nostalgic, homesick, wistful, maudlin, regretful |
| jealous | jealous, envious, covetous, envy |
| anticipating | anticipating, prediction, prevision, expect, prescience, expectation |
| embarrassed | embarrassed, awkward, cringe, abash, blush, discomfiture, chagrined |
| content | content, satisfy, gratify, meet, satisfied |
| devastated | devastated, demolish, destroy, ravage, raze, ruin, wreck, desecrate, desolate, despoil |
| sentimental | sentimental, slushy, maudlin, bathetic, mushy, schmaltzy, soppy |
| caring | caring, lovingness, loving, care, affectionate, sympathetic |
| trusting | trusting, trust, believable, believe, trusty |
| ashamed | ashamed, mortified, humiliated, abashed |
| apprehensive | apprehensive, perturbed, troubled, agitated, uneasy, perturbed |
| faithful | faithful, loyal, reliable, staunch, fidelity, honest, allegiance, faith |

Table 12: The emotion-related words of all 32 emotions for dataset EMPATHETICDIALOGUES.

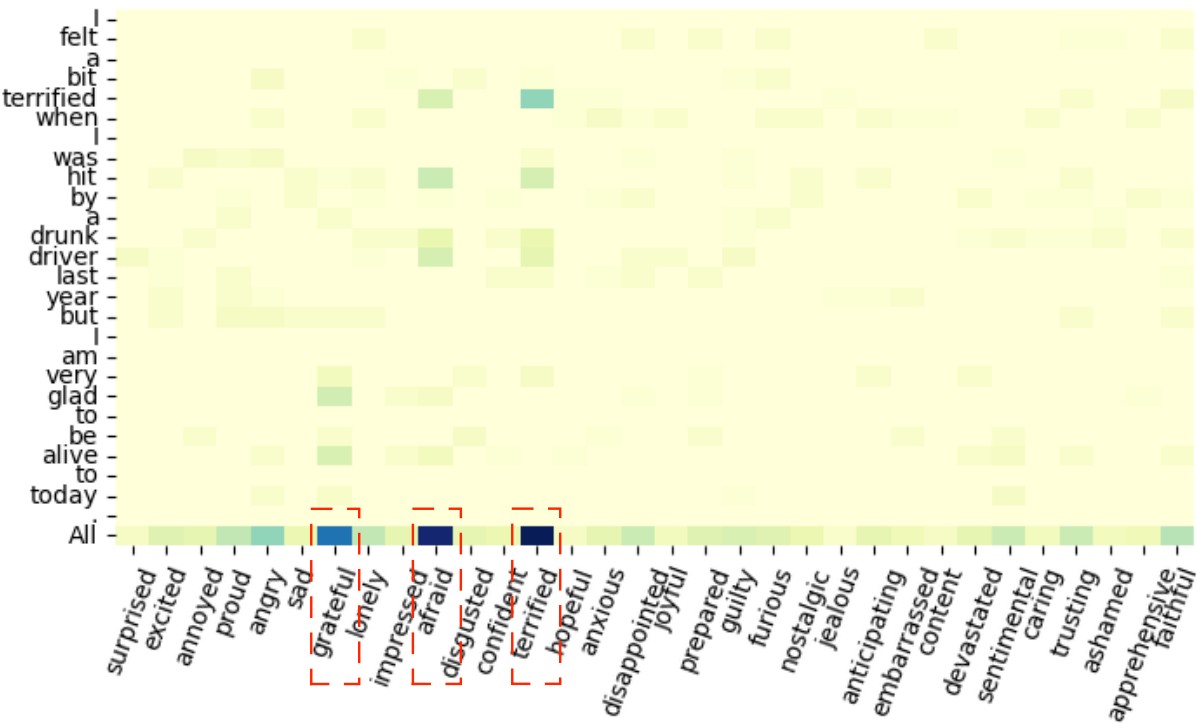

Figure 5: The visualization of the fusion of word-to-emotion edge features for E-CORE.

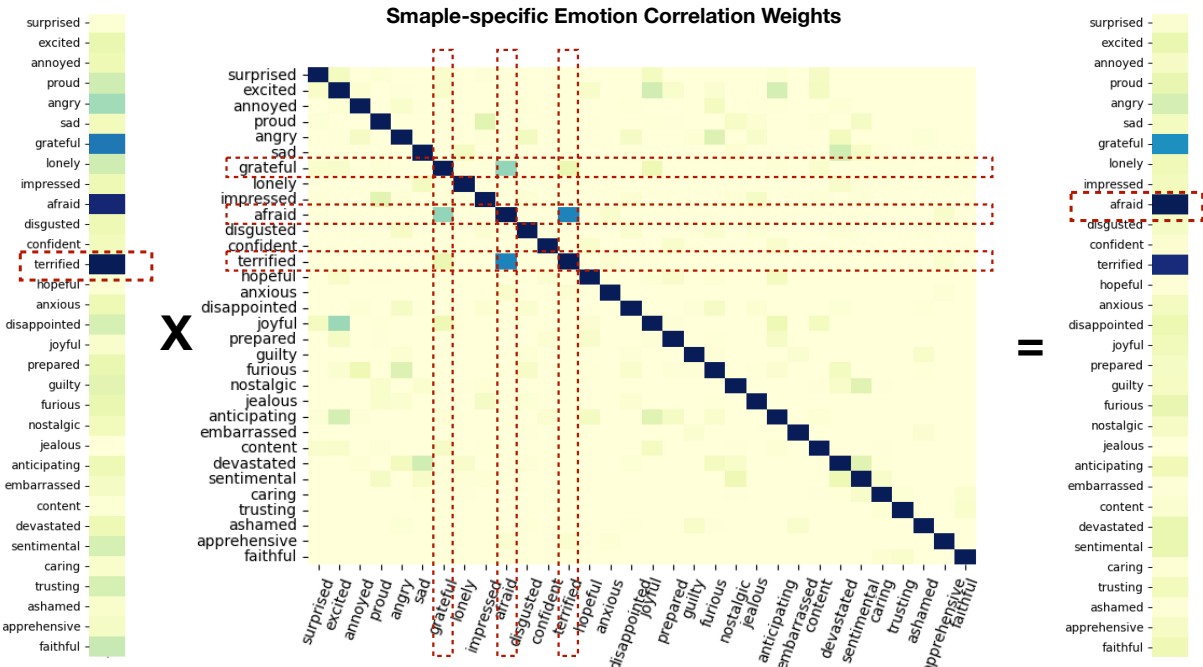

Figure 6: The visualization of the correlation-aware aggregation for emotion features in E-CORE.