# OpenReview forum: "E-CORE: Emotion Correlation Enhanced Empathetic Dialogue Generation"
_EMNLP/2023/Conference — EMNLP 2023 Main_

### Official Review · Reviewer_rPyr · 2023-08-04

**Soundness:** 4

**Excitement:**

3: Ambivalent: It has merits (e.g., it reports state-of-the-art results, the idea is nice), but there are key weaknesses (e.g., it describes incremental work), and it can significantly benefit from another round of revision. However, I won't object to accepting it if my co-reviewers champion it.

**Paper Topic And Main Contributions:**

A novel empathetic dialogue generation framework based on emotion correlation enhancement is proposed in this paper. A multi-resolution emotion graph is constructed to capture subtle emotion interactions, and an emotion enhanced decoder is devised for emotion perception and response generation.

**Questions For The Authors:**

1) It’s not clear whether an emotion can be deemed as significant if its original score is not high. The original score of “afraid” in Fig. 1 is lower than that of “terrified”. It puzzles me how to make “afraid” surpass “terrified” after combining with a correlation weight matrix. Additionally, how to capture the “afraid” emotion in this dialogue? It seems no emotion-related words regarding “afraid” can be found.

2) Why to model emotion correlations in a directed graph? Emotions are mutually related, and it’s hard for me to comprehend the afraid-to-terrified weight is different from the terrified-to-afraid weight.

3) “Sorry to hear” is claimed as an unsuitable response in Fig. 1, while it is also generated by the proposed method.

4) How to initialize the re-parameter matrix S?

5) Some notations have not been formally introduced, e.g., e_w, e_p, and e_d in Eq. (1), A_ij and E_ij in Eq. (7), etc.

6) In human evaluation, it’s unclear why E-CORE can achieve best under the metrics of fluency and relevance. It seems E-CORE pays less attention on fluency and relevancy learning in its response generation.

**Reasons To Accept:**

1) The paper is mostly well organized and easy to follow;

2) The study is interesting and may inspire the dialogue generation field;

3) The visual demonstrations are impressive, as well as the intuitive examples in Table 9 and Table 10.

**Reasons To Reject:**

1) The motivation seems not clear;

2) Mathematical expressions should be more rigorous;

3) Some experimental results have not been well explained.

**Reproducibility:**

3: Could reproduce the results with some difficulty. The settings of parameters are underspecified or subjectively determined; the training/evaluation data are not widely available.

**Reviewer Confidence:**

4: Quite sure. I tried to check the important points carefully. It's unlikely, though conceivable, that I missed something that should affect my ratings.

---

> ### Author Rebuttal · Authors · 2023-08-29
>
> Dear Reviewer rPyr,
>
> Thanks for your constructive review. We are encouraged that you found our paper to be well organized and see the potential of our method in the dialogue generation field. We provide detailed answers to your comments next.
>
>
> ----
> > **R1:** The motivation seems not clear.
>
> Thank you for your concerns. The motivation is that existing studies ignore the intrinsic emotion correlation in dialogues, resulting in inaccurate emotion perception and inappropriate response generation (details in questions 1 and 3). As the psychology studies suggest an intrinsic correlation in human emotions, we propose to break the emotion independence assumption existing in current methods, and explicitly mine and incorporate the emotion correlation to enhance emotion perception and response generation.
>
> ----
> > **R2:** Mathematical expressions should be more rigorous;
> > **& Q5:** Some notations have not been formally introduced, e.g., e_w, e_p, and e_d in Eq. (1), A_ij and E_ij in Eq. (7), etc.
>
> We will carefully check all mathematical expressions as you suggested, to ensure the rigor of each expression and notation.
> And the notations in Eq. (1) and (7) have been introduced. In Eq. (1), e_w, e_p, and e_d respectively indicate three kinds of embeddings: word embedding, position embedding and dialog state embedding, as stated in lines 204-206. In Eq. (7), A_ ij indicates the calculated attention score of node i to neighboring node j (defined in Eq. (6)), E_ ij indicates the edge weight from node i to node j (the initialization process is defined in Eq.(3), and the update process is defined in Eq.(5)).
> We will put these definitions closer to the equations, to make the expressions clearer.
>
>
> ----
> > **R3:** Some experimental results have not been well explained;
> > **& Q6:** In human evaluation, it’s unclear why E-CORE can achieve best under the metrics of fluency and relevance. It seems E-CORE pays less attention on fluency and relevancy learning in its response generation.
>
> We will first elaborate on the conclusions supported by existing experimental results. With the comparisons with SOTAs, we have demonstrated the promise of emotion correlation learning in both emotion perception (8.34% in Acc) and empathetic generation (8.53% in PPL,16.7% in Dist-2). With the ablation studies and visualization analysis, we further validated the essence of emotion correlation learning in our method E-CORE.
>
> Following your suggestions, we will further check all experiments and analyses to improve the explanation. Specific:
> **For the fluency metric**, all models show great performance and our model only make a  slight improvement compared to SOTAs, as the language models based on deep neural network are generally sufficient to generate fluent sentences.
> **For the relevancy metric**, our E-CORE significantly outperforms the SOTAs, as the emotion correlation learning helps provide more relevant emotions guidance, yielding empathetic responses rich in relevant emotions and semantics. The qualitative analyses (section 5.3) also confirm the conclusion, that our model generates more relevant and detailed phrases “good that you took them in”, “go back memories”, etc., while the SOTA models only produce universal responses.
>
> ----
> > **Q1** It’s not clear whether an emotion can be deemed as significant if its original score is not high. The original score of “afraid” in Fig. 1 is lower than that of “terrified”. It puzzles me how to make “afraid” surpass “terrified” after combining with a correlation weight matrix. Additionally, how to capture the “afraid” emotion in this dialogue? It seems no emotion-related words regarding “afraid” can be found.
>
> Thanks for this concern. We will first conduct a comprehensive analysis based on the visualization results Figure 5&6, and then answer your questions in turn.
>
> Firstly, the model captured the original dialogue emotion scores by fusing the attention weights of all dialogue words to each emotion, among which the words “terrified”, “hit” and “drunk driver” contribute to emotions “terrified” and “afraid”, and the words “glad” and “alive” contribute to “grateful” (Figure 5). Then a correlation-aware reweighting and aggregation is conducted on original scores, to obtain the final emotion scores. With a greater correlation weight of emotion “grateful” for “afraid” than for “terrified”, “afraid” showed greater score than “terrified” after aggregation (Figure 6). Thus:
>
> 1)**Whether an emotion is deemed as significant does not only depend on its original score.** As the final emotion scores are determined by both original score and emotion correlation weights, **an emotion with a low original score but strongly correlated to other high-score emotions, may also have higher final scores.** (For example, in the case of case study, emotion “nostalgic” is deemed as significant, due to the strong correlation between “sentimental” and “nostalgic”)
>
> 2)As in Figure 6, **“afraid” surpasses “terrified”, due to the greater correlation weight of emotion “grateful” for afraid than for terrified.**
>
> 3)As in Figure 5, **the emotion “afraid” is captured with related words “terrified”, “hit” and “drunk driver”.**
>
> A detailed visualization analysis for the case of Figure 1 also has been provided in Appendix J. We will enhance the expression in the main text to provide clearer explanations.
>
> ----
> > **Q2** Why to model emotion correlations in a directed graph? Emotions are mutually related, and it’s hard for me to comprehend the afraid-to-terrified weight is different from the terrified-to-afraid weight.
>
> The emotion nodes are interconnected, with a symmetric matrix as the edge weights, so the emotion connection in the graph **is equivalent to an undirected graph**. The afraid-to-terrified weight **is the same as** the terrified-to-afraid weight.
> However, considering the directed connection of word nodes, the emotion graph is represented as a directed graph, for consistency in computation and representation.
> Thanks for the reminder, we will clarify the explanations to avoid misunderstandings.
>
> ----
> > **Q3** “Sorry to hear” is claimed as an unsuitable response in Fig. 1, while it is also generated by the proposed method.
>
> In the case of Figure 1, the speaker first expressed the **afraid for accident**, and then transited to the **grateful for survival**.
> For empathetic response generation, a suitable response should respond to the transited emotion, which is more in line with the speaker's expectations and communication habits.
> Thus, we explain that only replying with 'sorry to hear' is not empathetic enough, while our proposed method replied both 'sorry to hear' and 'good that you look them in', which is a comprehensive response for the accident (afraid ) and survival (grateful).
> Thanks for the reminder. We will further strengthen the expression to clarify the difference between the responses to avoid misunderstandings.
>
>
> ----
> > **Q4** How to initialize the re-parameter matrix S?
>
> The re-parameter matrix S is a learnable parameter that is **randomly** initialized during the initial training process, and then continuously updated along with model training.

---

### Official Review · Reviewer_9WcU · 2023-08-04

**Typos Grammar Style And Presentation Improvements:** N/A
**Soundness:** 4

**Excitement:**

4: Strong: This paper deepens the understanding of some phenomenon or lowers the barriers to an existing research direction.

**Missing References:**

N/A

**Paper Topic And Main Contributions:**

Previous works adopt an independent assumption on different emotions by first predicting emotion labels and then generating responses. This paper considers the co-occurrence of multiple emotions and proposes to mine and incorporate this intrinsic emotion correlation into single-labeled empathetic dialogue generation datasets.

**Questions For The Authors:**

1.	Are all emotion nodes connected to each other for the graphs of all instances?
2.	I like the motivation of emotion correlation, but the implementation of the overall method is a little bit complicated. Do you compare your method with directly few-shot prompting LLMs? What are the advantages of your methods and what are the drawbacks of the current LLMs when performing empathetic response generation?


**Reasons To Accept:**

1.	The authors propose a very new perspective to break the emotion independence assumption existing in current empathetic dialogue generation methods and model the intrinsic emotion correlation.
2.	The experimental results show the usefulness of the proposed method in different evaluation methods.


**Reasons To Reject:**

1.	It would be clearer to give a description of the overall process in the figure caption or the beginning of section 3 based on the contents of Figure 3.
2.	Are all emotion nodes connected to each other for the graphs of all instances?
3.	I like the motivation of emotion correlation, but the implementation of the overall method is a little bit complicated. Do you compare your method with directly few-shot prompting LLMs? What are the advantages of your methods and what are the drawbacks of the current LLMs when performing empathetic response generation?


**Reproducibility:**

4: Could mostly reproduce the results, but there may be some variation because of sample variance or minor variations in their interpretation of the protocol or method.

**Reviewer Confidence:**

4: Quite sure. I tried to check the important points carefully. It's unlikely, though conceivable, that I missed something that should affect my ratings.

---

> ### Author Rebuttal · Authors · 2023-08-29
>
> Dear Reviewer 9WcU,
>
> Thanks for your insightful review. We are excited that you appreciated the contribution and the novelty of our work. We improve the expression as you suggested to improve clarity, and provide a detailed discussion on your questions.
>
> ----
> > **R1:** It would be clearer to give a description of the overall process in the figure caption or the beginning of section 3 based on the contents of Figure 3.
>
> Thanks for this constructive suggestion. We will supplement a description of the overall process in the figure caption of Figure 3. The details are as follows:
> The overall process consists of 3 phases: 1)context encoding: to encode the dialogue context and all emotions into embedding features and contextual representation; 2)multi-resolution emotion graph network: to capture the context-based emotion interaction from different resolutions, further encoding the emotion correlation; 3)emotion correlation enhanced decoding: to incorporate the emotion correlation for enhancing emotion signal perception and response generation, respectively.
>
> ----
> > **R2&Q1:** Are all emotion nodes connected to each other for the graphs of all instances?
>
> Thank you for your concerns. The emotion nodes are **connected to each other in graph modeling, but may not be the case in decoding phase**.
> In the multi-resolution emotion graph modeling phase, all emotion nodes **are connected to each other for the graphs of all instances**, to model the emotion correlation (Section 3.2).
> In the emotion correlation enhanced decoding phase, when using the  hard strategy, the soft/hard gated generator improves the emotion graph by removing the context-irrelevant emotion nodes, so the emotion nodes **are not connected to each other** (Section 3.3 and Figure 3-b).
> We will improve the expressions to provide clearer explanations.
>
> ----
> > **R3&Q2:** I like the motivation of emotion correlation, but the implementation of the overall method is a little bit complicated. Do you compare your method with directly few-shot prompting LLMs? What are the advantages of your methods and what are the drawbacks of the current LLMs when performing empathetic response generation?
>
> Thanks for your constructive suggestions. We **have compared our model combined with full pre-trained LLM (DialoGPT [1])** in Appendix E, but not compared it with directly few-shot prompting LLMs, as we think full pre-trained LLM exhibits better performance than the few-shot one, so the comparison is conducted with the better one.
> According to your suggestions, **we conducted a detailed comparison of our model and directly few-shot prompting LLMs (DialoGPT [1], ChatGPT [2]). As shown in Table 1, Our model outperforms DialoGPT in all performance, but still falls short of ChatGPT in response quality**. We will further **conduct a detailed analysis** of the comparison and **then summarize the advantages** of our model.
>
> Table 1: Performance comparison with few-shot prompting LLMs. Using metric BLEU instead of PPL to evaluate the response quality, as LLMs directly generate results rather than probability distributions, unable to calculate PPL.
> | Model            |BLEU@1| BLEU@4    |   Acc   |    Dist-1      |      Dist-2      |
> |------------------|:-----------:|:-----------:|:-----------:|:-----------:|:-----------:|
> | DialoGPT  [1]   | 33.5 |11.4|  30.59   | 0.46  |    3.02  |
> | ChatGPT  [2]   |**45.4**| **14.5**|  35.71 |  0.58 |   3.34   |
> | **Ours-Soft**  |43.8|13.7| 42.57|0.68 | 3.38 |
> | **Ours-Hard**  |44.2|14.2| **42.59**|**0.72** | **3.49** |
>
> **1). Result analysis:**
> The results are shown in Table 1. Our model outperforms DialoGPT in all performance, with higher emotion accuracy than ChatGPT, but still falls short of ChatGPT in response quality, which is basically consistent with the results of manual evaluation. It should be noted that the metric BLEU doesn’t fully reflect the gap between our model and ChatGPT in generation quality, as ChatGPT without fine-tuning generates high-quality responses but not similar to ground-truth responses. This also indicates that current evaluation metrics based on ground-truth comparison, cannot objectively measure the effectiveness of LLMs.
>
> **2). Summary of Advantages of our model:**
>
> Table 2: Comparisons of parameter quantity and generation speed (per sample) with few-shot prompting LLMs.
> | Model            |    parameters    | generation speed      |
> |------------------|:-----------:|:-----------:|
> | DialoGPT  [1]  |   345M  |0.675s   |
> | ChatGPT  [2]  |175B |1.328s|
> | **Ours-Soft**  |**36.1M** |**0.034s**|
> | **Ours-Hard** | **36.1M** |0.036s|
>
> **<1>Fewer parameters and higher generation speed**: As shown in Table 2, our model achieves better performance than DialoGPT with  1/10 parameters (36.1M vs 345M), only inferior to ChatGPT (175B). For the generation speed, our model respectively achieves 19.8x and 39.1x speedup compared to online DialoGPT and ChatGPT.
>
> **<2>More accurate emotion prediction**: Our model achieves a more accurate emotion prediction compared to ChatGPT and DialoGPT (Acc: 42.59% vs 35.71%&30.59%). We observe that LLMs as the universal models, are inadequate in distinguishing similar emotions such as angry and annoyed. And our model with emotion correlation learning is good at the fine-grained emotion perception in the empathetic dialogue generation task.
>
> **<3>Strong scalability**: Our model has stronger scalability, as can be combined with other pre-trained models for joint training to achieve better performance. As in Tabel 6, our model trained with pre-trained language model DialoGPT shows excellent response performance (BLEU-1:0.624, Dist-1:3.07 Dist-2:4.92) and emotion accuracy (Acc: 50.12%).
>
> We will supplement all the results and analysis to the revision as an extension.
>
> ----
> **Attached the prompt used for few-shot prompting LLMs comparison:**
>
> 1）For response generation:
>
> Your task is to generate a response in a consistent way.\
> <context>: S1:I went through some of my old stuff yesterday, and I found my security blanket that I used when I was a kid!\
> <response>: Awww I bet that brought back memories.\
> <context>: S1:My husband is the most faithful man. L1:That is great to hear! A faithful spouse is a blessing.S2:I have so many health problems and he is always there for many no matter what being loving and caring.\
> <response>: I am sorry to hear about that! I hope everything gets better for you!\
> <context>: xxxx
>
> 2）For emotion perception:
>
> Please choose one words from the follwing set to describe above sentence in a consistent way. set: surprised, excited, annoyed, proud, angry, sad, grateful, lonely, impressed, afraid, disgusted, confident, terrified, hopeful, anxious, disappointed, joyful, prepared, guilty, furious, nostalgic, jealous, anticipating, embarrassed, content, devastated, sentimental, caring, trusting, ashamed, apprehensive, faithful. \
> <context>: S1:I went through some of my old stuff yesterday, and I found my security blanket that I used when I was a kid!\
> <emotion>: sentimental\
> <context>: S1:My husband is the most faithful man. L1:That is great to hear! A faithful spouse is a blessing.S2:I have so many health problems and he is always there for many no matter what being loving and caring.\
> <emotion>: faithful\
> <context>: xxxx
>
>
>
> **Reference:**
>
> [1] Dialogpt: Large-scale generative pre-training for conversational response generation, ACL, 2020.
>
> [2] https://openai.com/blog/chatgpt. Chatgpt, OpenAI, 2022.

---

### Official Review · Reviewer_9bdy · 2023-08-11

**Soundness:** 4

**Excitement:**

3: Ambivalent: It has merits (e.g., it reports state-of-the-art results, the idea is nice), but there are key weaknesses (e.g., it describes incremental work), and it can significantly benefit from another round of revision. However, I won't object to accepting it if my co-reviewers champion it.

**Missing References:**

Wang L, Li J, Lin Z, et al. Empathetic Dialogue Generation via Sensitive Emotion Recognition and Sensible Knowledge Selection[C]//Findings of the Association for Computational Linguistics: EMNLP 2022. 2022: 4634-4645.

**Paper Topic And Main Contributions:**

Existing studies ignore the intrinsic emotion correlation in dialogues, resulting in inaccurate emotion perception and unsuitable response generation. Therefore, the paper propose a novel emotion correlation enhanced empathetic dialogue generation framework, which capture multi-resolution emotional interactions through dialogue history contexts and eventually complete empathic responses.

**Reasons To Accept:**

1. The paper is well experimented and demonstrates meaningful enhancement of the experimental results through the use of advanced large-scale modeling and visualization and other work;
2. The study is meaningful and innovative in improving the quality of dialog generation by introducing affective relevance;
3. The paper is easy to understand and beautifully laid out.

**Reasons To Reject:**

1.In the introduction and the experimental analysis, the authors marked the magnitude of the metrics enhancement（8.53% in PPL，16.7% in Dist-2，8.34% in Acc），It is suggested that the change be made to the absolute value of the enhancement rather than the percentage to reduce ambiguity;
2.The performance of emotion intensity labeling has an impact on the subsequent calculation of sentiment correlations was not clarified in the paper (e.g., the relationship between the loss generated when the model is labeled with emotion intensity and the overall loss of the model proposed by the authors);
3. The paper mentions that emotion graphs require modeling the correlation of intrinsic emotions and interconnecting all emotion nodes, and that the details of the main emotion and secondary emotions conversions should be detailed when calculating the emotion correlations and subsequent generation;
4. The authors mention in the experimental section that their proposed model is compared to the Sotas model. As far as I know, the Baselines chosen for the paper is not the SOTA model. The authors can be confirmed by the following literature.
Empathetic Dialogue Generation via Sensitive Emotion Recognition and Sensible Knowledge Selection

**Reproducibility:**

4: Could mostly reproduce the results, but there may be some variation because of sample variance or minor variations in their interpretation of the protocol or method.

**Reviewer Confidence:**

4: Quite sure. I tried to check the important points carefully. It's unlikely, though conceivable, that I missed something that should affect my ratings.

---

> ### Author Rebuttal · Authors · 2023-08-29
>
> Dear Reviewer 9bdy,
>
> Thank you for your valuable review. We are encouraged that you found our approach novel and technically sound. We provide detailed answers to your comments below. Hope to address your concerns.
>
> ----
> > **R1:** In the introduction and the experimental analysis, the authors marked the magnitude of the metrics enhancement（8.53% in PPL，16.7% in Dist-2，8.34% in Acc），It is suggested that the change be made to the absolute value of the enhancement rather than the percentage to reduce ambiguity.
>
> Following your suggestion, we will provide both the absolute values and percentages of the enhancement, where absolute values more directly reflect the model's breakthroughs in various metrics on response quality and emotion accuracy (3.08 in PPL, 0.5 in Dist-2, 3.28 in Acc), and percentages more intuitively reflect the improvement of our model compared with SOTAs (8.53% in PPL, 16.7% in Dist-2, 8.34% in Acc).
> Thanks for your valuable suggestion. We will provide absolute values as the main explanation for performance improvement, with percentages as additional, and provide clear explanations for both to avoid misunderstandings.
>
>
> ----
> > **R2:** The performance of emotion intensity labeling has an impact on the subsequent calculation of sentiment correlations was not clarified in the paper (e.g., the relationship between the loss generated when the model is labeled with emotion intensity and the overall loss of the model proposed by the authors).
>
> The emotion intensity **indirectly impacts** the calculation of emotion correlation **by influencing the graph construction**. Specifically, emotion intensity indicates the emotional importance of dialogue words, guiding the word connections at different resolutions and serving as initial edge weights, enabling the emotion graph to capture the emotion interactions from multi-resolution, further encoding the emotion correlation. Words with higher emotion intensities tend to have a greater effect on the emotion graph, thus playing a greater role in the subsequent calculation of emotion correlations.
>
> For the case you provided, the loss generated when the model is labeled with emotion intensity and the overall loss of the model should be the same, as the model is already labeled with emotion intensity. To avoid not fully understanding your question, we **supplement all ablation experiments on the performance of emotional intensity**, by comparing the variants without emotion intensity or with different emotion intensity labeling on both soft and hard strategies. The results are shown in Table 1 below.
>
> Table 1: Ablation studies on the performance of emotion intensity based on both soft and hard strategies. [1,2,3] are other emotion lexicons used to provide different emotional intensity labeling, and [4] is used in the original paper.
>
> | Model            |   PPL         |  Acc  |      Dist-1      |      Dist-2      |
> |------------------|:-----------:|:-----------:|:-----------:|:-----------:|
> | Soft-w/o       |  34.97   |  40.06   | 0.54  |   3.02   |
> | Soft-SentiWordNet [1]         |   33.46   |42.05 |   0.63 |     3.34  |
> | Soft-VADER  [2]      |    33.35   | 42.17  |     0.65  |   3.32   |
> | Soft-VAD  [3]        |   33.39  |  42.28   |     0.66  |    3.36  |
> | **Soft-SKEP [4] (Ours)** | **33.04** |**42.57**|  **0.68** | **3.38** |
> | Hard-w/o       |  34.91   | 40.12   |  0.57  |   3.06   |
> | Hard-SentiWordNet [1]         |   33.42   |42.02 |   0.65 |     3.38  |
> | Hard-VADER  [2]      |    33.32   |42.07  |      0.66  |   3.42   |
> | Hard-VAD  [3]        |   33.35  |  42.35   |   0.68  |    3.44  |
> | **Hard-SKEP [4] (Ours)** | **33.03** |  **42.59**| **0.72** | **3.49** |
>
> The results show that: 1) model without emotion intensity, which degenerates into a single-resolution emotion graph, performs weakly, indicating the great impact of multi-resolution graph modeling for correlation learning. 2) **model exhibits strong robustness to different emotion intensity labeling**, also indicating that the emotion correlation learning more relies on graph training, instead of the performance of the original emotion intensity.
> If there are any questions for you, we welcome more specific explanations in the subsequent discussion stage and will actively respond.
>
>
>
> ----
> > **R3:** The paper mentions that emotion graphs require modeling the correlation of intrinsic emotions and interconnecting all emotion nodes, and that the details of the main emotion and secondary emotions conversions should be detailed when calculating the emotion correlations and subsequent generation.
>
> We would clarify that the **main emotion and secondary emotions conversions do not explicitly occur in the calculating of emotion correlations and subsequent generation, but occur in emotion signal perception**, as stated in Section 3.3. Specifically, for correlations calculating, all emotions are uniformly treated; and for subsequent generation, main emotion and secondary emotions are all considered as co-occurrence emotions to guide generation. So in these phases, emotions conversions do not occur.
>
> Thanks for your suggestion. We will **provide more detailed explanations for the emotions conversions** in the emotion signal perception. Specifically, the model first obtains the original dialogue emotion scores by fusing the attention features of dialogue words to each emotion, then conducts the correlation-aware aggregation to obtain the final emotion scores. We provide a more intuitive explanation based on the case in Figure 1. Based on the original emotion scores, “terrified” is the main emotion and “afraid” “grateful” are the secondary emotions. After correlation-based reweighting and aggregation, due to the greater correlation weight from “grateful” to “afraid” than to “terrified”, "afraid" surpasses "terrified" to become the main emotion (Figure 6).
>
> In addition, we also conducted experiments to calculate the proportion of the conversions between main emotion and secondary emotions (i.e., the proportion of main emotion changes before and after correlation-aware aggregation), as well as the absolute improvement it brings to emotion accuracy, as shown in Table 2.
>
>
> Table 2: The results of the proportion of emotions conversions and its absolute improvement in emotion accuracy.
>
> | Models | conversions proportion | Absolute Improvement in Acc |
> | -------- | :--------: | :--------: |
> | **Ours-Soft**    | 25.88%     | 3.31     |
> | **Ours-Hard**    | 15.09%     | 2.61     |
>
> The results indicate that our correlation-aware aggregation strategy helps achieve a conversion from co-occurrence emotions (including main and secondary) to main emotion, which effectively improves the accuracy of main emotion prediction.
>
>
> ----
> > **R4: The authors mention in the experimental section that their proposed model is compared to the Sotas model. As far as I know, the Baselines chosen for the paper is not the SOTA model. The authors can be confirmed by the following literature. Empathetic Dialogue Generation via Sensitive Emotion Recognition and Sensible Knowledge Selection.**
>
> Thanks for your suggestion. We have added a comparison for the work [5], as shown in Table 3 below. It is worth noting that [5] uses additional utterance-level emotion annotations for fine-grained emotion encoding, while our model only uses dialogue-level emotion annotations. Therefore, we would **explain from two settings: direct comparison with the SOTA [5], and the comparison with the SOTA [5] under same annotations, to clarify the advantages of our model compared to [5].**
>
> 1). direct comparison with the SOTA [5]: **our model achieves the best performances in the most important metrics PPL and Acc, and near performances in Dist (-0.01 in Dist-1, +0.26 in Dict-2).** As PPL and Acc measure the consistency between generated responses/emotions and real responses/emotions, compared with Dist which measures the generation diversity, they can more directly and objectively reflect the model's ability to generate a relevant and empathetic dialogue.  **These results fully demonstrate that our model has achieved usually better response quality and empathetic perception for the Empathetic dialogue generation task**, even in a lack of information compared to [5], also achieves near performance in generation diversity.
>
> 2). comparison with the SOTA [5] under same annotations: i.e., comparing with the SOTA(w/o Utter) model below, **our model outperforms the SOTA in all metrics, and exhibits a significant improvement in PPL(4.34 in absolute, 11.6% in relative) and Acc (3.69 in absolute, 9.5% in relative).** These results demonstrate the superiority of our model in empathic understanding and expression, when using the same input information.
>
>
> Table 3: Performance comparison with the SOTA model.
> | Model            |    PPL    | Acc      |      Dist-1      |      Dist-2      |
> |------------------|:-----------:|:-----------:|:-----------:|:-----------:|
> | SOTA [5]    |37.03 |  41.85   | **0.73**  |   3.23   |
> | SOTA (w/o Utter) [5]   |37.37 |  38.90   | 0.70  |   3.13   |
> | **Ours-Soft**       |   33.04 |    42.57  |   0.68  |    3.38     |
> | **Ours-Hard** | **33.03** | **42.59**|0.72 | **3.49** |
>
> Following your suggestion, we also added a detailed discussion on work [5]. This paper [5] proposed a serial encoding and emotion-knowledge interaction method for the task, effectively utilizing fine-grained emotion features and commonsense knowledge to enhance empathy response. As our method is distinct from [5], by modeling and utilizing the emotion correlation to achieve enhancement, we believe these two methods can be complementary to each other.
> We will provide a detailed explanation and comparison of this work in the related work and experimental sections in the revision. Thank you for guiding us toward a more accurate discourse in our revised manuscript.
>
>
>
>
>
>
>
>
> **Reference:**
>
> [1] SENTIWORDNET: A publicly available lexical resource for opinion mining, ELRA, 2006.
>
> [2] Vader: A parsimonious rule-based model for sentiment analysis of social media text, AAAI, 2014.
>
> [3] Knowledge-Enriched Transformer for Emotion Detection in Textual Conversations, EMNLP-IJCNLP, 2019.
>
> [4] Skep: Sentiment knowledge enhanced pre-training for sentiment analysis, ACL, 2020.
>
> [5] Empathetic Dialogue Generation via Sensitive Emotion Recognition and Sensible Knowledge Selection, EMNLP, 2022.

---

### Meta-Review · Area_Chair_rcrw · 2023-09-19

**Recommendation:** 4

**Metareview:**

All reviewers appreciate the soundness of the work although they disagree about excitement, I think the question really revolves around the impact of modeling emotionally correlation in dialogue. I think this speaks to rPyr point about motivation.

It is somewhat concerning that the details of the human evaluations are not stated in the appendix nor do there seem to be tests of statistical significance. Given the large superiority I do not see this is a fatal flaw, but something that the authors must include.

---

### Decision · Program_Chairs · 2023-10-07

**Decision:**

Accept-Main

**Comment:**

All reviewers appreciate the soundness of the work although they disagree about excitement, I think the question really revolves around the impact of modeling emotionally correlation in dialogue. I think this speaks to rPyr point about motivation.

It is somewhat concerning that the details of the human evaluations are not stated in the appendix nor do there seem to be tests of statistical significance. Given the large superiority I do not see this is a fatal flaw, but something that the authors must include.